# Pediatric Neurology Workforce in Saudi Arabia: A 5-Year Update

**DOI:** 10.3390/healthcare11162288

**Published:** 2023-08-14

**Authors:** Ahmed K. Bamaga, Anas S. Alyazidi, Albatool Almubarak, Mohammad N. Almohammal, Ayidh S. Alharthi, Matar A. Alsehemi

**Affiliations:** 1Department of Pediatrics, Faculty of Medicine, King Abdulaziz University, Jeddah 21589, Saudi Arabia; abamaga@kau.edu.sa; 2Neuromuscular Medicine Unit, Department of Pediatrics, Faculty of Medicine, King Abdulaziz University Hospital, Jeddah 21589, Saudi Arabia; 3Faculty of Psychology, University of Edinburgh, Edinburgh EH10 5HF, UK; almubarak.batool@gmail.com; 4Department of Pediatrics, Ministry of Health, Bisha 67841, Saudi Arabia; dr.mohammad.ns@gmail.com; 5Pediatric Neurology Unit, Department of Pediatrics, King Fahad Hospital, Albaha 65732, Saudi Arabia; mr_ayed_x5@hotmail.com (A.S.A.); matar5645@gmail.com (M.A.A.)

**Keywords:** pediatric neurology, workforce, practice, resident training, Saudi Arabia

## Abstract

Background: The medical workforce plays a pivotal role in advancing human health, particularly within the healthcare system of Saudi Arabia. While government-employed healthcare providers form the central structure of the system and offer free healthcare services, the private healthcare sector is also witnessing significant growth. In parallel, the field of child neurology has experienced notable transformations in recent years, with continued expansion. This expansion brings forth a range of challenges for both current and future pediatric neurologists, necessitating careful consideration and proactive measures to address them. Aim of the study: To investigate and analyze the current characteristics of the workforce, with a specific focus on their employment status and related data. Methods: This is a cross-sectional analysis, using a survey to assess the distribution of pediatric neurologists in Saudi Arabia (SA). The final analytical sample included 82 subjects, working in 13 regions in SA. A descriptive analysis was used to address the study question. Results: The survey received responses from a total of 82 pediatric neurologists in Saudi Arabia (response rate 55%), with 38 (46%) being men and 44 (54%) being women. The mean age was 33 ± 1.225 years. The majority of participants practiced in major cities such as Riyadh and Jeddah. Nearly 50% of pediatric neurologists experienced some form of delay in obtaining their first job, ranging from 1 to 36 months. Conclusion: The landscape of the pediatric neurology workforce is currently witnessing noteworthy demographic shifts. With the majority of practitioners concentrated in major cities, there is an ongoing demand for qualified professionals in peripheral areas. This study describes the real-life challenges faced by pediatric neurologists, particularly the delay in securing employment after graduation, and underscores the critical importance of addressing these persistent issues along the journey of pediatric neurology.

## 1. Introduction

The medical workforce is a key pillar in advancing human health; it is an essential element that enables healthcare systems to function adequately [1]. In Saudi Arabia, the central structure of the healthcare system is based upon free healthcare services via publicly owned facilities and government-employed healthcare providers [2]. The number of healthcare facilities in Saudi Arabia continues to increase with the growing population and the increasing need for healthcare services [2]. Continued and expanded efforts have been made by Saudi Arabia since the late 1950s to empower, enable, and enroll local physicians into the workforce, with local and international training opportunities being offered to healthcare providers [2]. Those healthcare providers were considered a cornerstone of the healthcare system. Studies conducted on the Saudi population have uncovered a significant annual population growth rate, ranging from approximately 3% to 3.5% in 1994 [3,4]. However, recent data indicate a further increase in the growth rate to 4.5% [5], accompanied by a total fertility rate of 7.1% [4]. These findings, coupled with previous research that established chronic neurologic disorders as the most prevalent type of disorders among the pediatric population [6], underscore the importance of comprehending the landscape of pediatric neurology and its workforce. Large studies that screened thousands of children in Saudi Arabia emphasized the increasing prevalence of specific chronic neurological disorders, i.e., above average estimates; these included cerebral palsy, mental retardation, epilepsy, hydrocephalus, neural tube defect, as well as for other neurological disabilities [6]. With this increase in the prevalence of such conditions, the child neurology field has undergone great transformation in the past few years, and it continues to grow and expand to meet the increasing demand. The expansion of residency programs and the establishment of new training centers throughout the country have contributed to a growing number of pediatric neurologists entering the workforce. However, this growth also brings about a range of issues and challenges that necessitate a thorough and rational assessment. Natural questions to ask are whether the workforce is equipped to handle these challenges. One key challenge is the current number of pediatric neurologists and the future need. The only available nationwide studies, which were published in 2005 and 2017, concluded that the total number of pediatric neurologists per 100,000 children remains well below the standardized international ratio (0.4:100,000) with even greater shortages in some regions despite significant improvements, as observed by a recent study [7,8]. Since then, the literature has described a lack of updates on the reality of the Saudi pediatric neurology workforce. On the other hand, and as the workforce expands, more challenges arise, such as the occurrence of delays in the employment of recently trained pediatric neurologists. This study aims to investigate and analyze the current characteristics of the workforce, with a specific focus on their employment status and related data.

## 2. Materials and Methods

### 2.1. Participants and Procedure

This is a cross-sectional analysis, using a survey to present the distribution of pediatric neurologists in Saudi Arabia (SA). The study was conducted in May 2023 in accordance with the Strengthening the Reporting of Observational Studies in Epidemiology (STROBE) reporting guidelines for cross-sectional research [9]. The study included 82 pediatric neurology residents and graduates across the 13 regions of Saudi Arabia. Residents and graduates were identified through national databases (i.e., the Saudi Pediatric Neurology Society). Participants were enrolled using an electronic, self-administered questionnaire using Google Form, a freely-available online questionnaire tool. The self-administered questionnaire was distributed using the WhatsApp instant messaging application and email services. Pediatric neurology consultants and specialists were excluded from the study.

### 2.2. Survey Design

A 26-item questionnaire, created according to the study objectives, with a few items adopted from previous literature [8], was utilized in this study. The questionnaire consisted of four sections. The first section included a consent statement, whereby the study participants acknowledged that their information would be utilized for research purposes and confirmed that they were part of the target population. No participant was allowed to progress before agreeing to this statement. The second section was intended to collect personal data; it identified participants’ age, gender, nationality, social status, residency program location, medical school location, type of their current training center (i.e., government hospital, academic institution, or private hospital), current year of residency, number of research publications, monthly income, and health insurance status. The third section was for current fellowship trainees. This included the type of current fellowship training and the training location. Section 4 included questions to assess subjects’ post-graduation status by asking about the timeframe required for individuals to enroll into the workforce, the city of their first job, the average number of patients per clinic, the type of available supporting services (i.e., epilepsy/EEG, clinical neurophysiology, neuromuscular/EMG, neurodevelopmental disabilities, sleep medicine, neuro-oncology, headache medicine, neurogenetics), if they have an academic job (i.e., lecturer, professor, etc.), if they have a research job (i.e., researcher, scientist, etc.), and finally, their current job title.

### 2.3. Data Analysis

Microsoft Excel version 20 was used to store the collected data. Statistical analyses were conducted using Rstudio (version 4.0.2). Categorical variables were the values of distinct groups based on a given set of characteristics; such variables were described in frequency tables, and continuous variables obtained by measuring were described with mean and standard deviation (SD) values. To assess whether participants in different employment groups differed in terms of demographic characteristics and variables of analytical interest, preliminary analyses were first conducted. Continuous variables were first assessed in terms of the normality of their distribution using the Shapiro Wilk test. If the normality assumption was met, a parametric two-sample *t*-test was conducted. If the normality assumption was not met, the Mann-Whitney U-test would be used. Categorical variables were compared between the two employment groups using Fisher’s exact test, given that the subsample of data was too small to conduct a Pearson’s Chi-squared test [10]. The significance of the effects of interest was estimated at a 95% confidence interval.

## 3. Results

### 3.1. Demographic Characteristics

A descriptive summary of demographic characteristics is shown in Table 1. A total of 82 pediatric neurologists in Saudi Arabia responded to the survey. The response rate was 55%. Participants enrolled in the study were deemed eligible by confirming that they were within the targeted population. Of the participants, 46% were men and 54% were women. Nearly all participants (99%) were Saudi nationals. The majority of participants were aged 32 years (n = 12), the mean age was 33.24 ± 5.66 years, and the age range was 25–62 years. As for participants’ relationship status, around 68% were married, 29% were single, and 2% were engaged.

The majority of participants (48%) were currently pursuing postgraduate studies, and nearly 20% were fresh graduates. The rest of the participants varied in their current training levels, whereby most (15%) described themselves as R3 trainees. Six participants described themselves as trainees currently in R5 and R4, with a prevalence of 7% for each group. Two participants described themselves as R1 trainees, while a single R2 trainee was identified. The majority (49%) of participants received a salary of >15,000 SAR; the rest fell between 12,000–15,000 (2%) and <1000 SAR (2%), and 35% decided to not answer the question. More than half of the participants (63%) did not receive health insurance.

### 3.2. Education, Research and Training

A descriptive summary of participants’ education, research, and training is shown in Table 2. Nearly all participants (98%) did their work residency in Saudi Arabia, with only two having done so abroad. Around 35% of participants did their residency at King Fahad Medical City (KFMC), 17% at King Faisal Specialist Hospital & Research Center (KFSHRC) in Riyadh, and nearly 15% at Prince Sultan Military Medical City (PSMMC). Figure 1 illustrates the locations of participants’ current workplace by state, indicating that most participants were clustered in Riyadh. Most participants (54%) started their work residency between 2014 and 2018, and most (54%) graduated between 2019 and 2023. As for participants’ undergraduate education, 17% studied at King Abdulaziz University (KAU), around 16% studied at King Saud University (KSU), and nearly 11% studied at King Khalid University (KKU). Most participants had one (23%), two (21%), or three (20%) research publications (Figure 2). Only 5% had published more than 10 papers. Around 29% of participants are currently enrolled on a fellowship program, mostly in Saudi Arabia (13%) or Canada (13%), and most (16%) are training in Epilepsy/EEG.

### 3.3. Employment Status

A descriptive summary of participants’ employment status is shown in Table 3. The majority of participants who received their first job in the same city they were born in were in Riyadh (18%); 5% who were born in Jeddah also received their first job there; 5% who were born in Alhassa also received their first job there; 4% who were born in Abha also received their first job there; 4% who were born in Al-Madinah also received their first job there; 4% who were born in Makkah also received their first job there; only 1% who were born in Arar also received their first job there; 1% who were born in Hail also received their first job there; 1% who were born in Tabuk also received their first job there; and 1% who were born in Al-Qassim also received their first job there.

More generally, and regardless of participants’ city of birth, nearly half of participants (49%) received their first job in Riyadh, and nearly 10% received their first job in Jeddah. As for participants’ location according to their first job, most participants were clustered in Riyadh, followed by Makkah. The majority of participants (46%) received their first job within 1–3 months from graduation; around 21% received their first job 4–6 months after graduation. Only around 3% waited 6–10 years before finding their first job. Figure 3 illustrates that most participants (24%) reported that the average number of patients they see per half-day session was between 6–10, followed by 20% who see an average of 11–15 patients, and 18% who see 21–25 patients per session. We removed a total of 10 observations from some specific city of birth and city of first job to maintain the privacy of respondents. 

### 3.4. Bivariate Analysis

Bivariate analyses between two employment groups on multiple characteristics are summarized in Table 4. Group 1 (n = 18) was defined as participants who were employed within their first three months after graduating from their pediatric neurology residency program; meanwhile, Group 2 (n = 21) included participants who were employed after at least three months of unemployment after graduating from a pediatric neurology residency program. The two employment groups did not significantly differ in age (*p* = 0.97), gender (*p* = 1.00), or city of birth (*p* = 0.13). However, the two groups significantly differed in their residency center (*p* = 0.04).

### 3.5. Supporting Services

Participants were surveyed on the available supporting services at their centers. EEG was the most widely available service, with 96.34% of respondents indicating the presence of EEG in their center. Physical therapy and social workers were reported to be available by 87.80% and 86.59% respondents, respectively. Other services, including a dietitian, psychology services, and speech and language pathology, were available in 81.71%, 75.61%, and 71.95% of centers. The percentages of respondents who indicated the presence of other services, i.e., a rehabilitation center, evoked response tests (i.e., VEP, BEAR, SSER), psychology services, occupational therapy services, physical therapy services, specialized nurses, an audiologist, and electromyography, are demonstrated in Figure 4.

## 4. Discussion

The aim of this study was to analyze the pediatric neurology workforce in Saudi Arabia in terms of demographics, educational and research activity and level, and employment status. We obtained data for 82 pediatric neurologists currently training and practicing in Saudi Arabia and abroad. This is the largest and most recent report among similar studies in Saudi Arabia. Furthermore, the study represents a contemporary update of previous studies [7,8] that analyzed the same workforce amid growing challenges and changes.

The significance of this study is underscored by the growing population and the subsequent increase in referrals and counseling demands placed on pediatric neurologists, as evidenced by prior investigations [11]. Furthermore, advancements in diagnostic modalities, particularly next-generation sequencing (NGS) that sequences DNA and RNA to detect pathogenic variants and gene mutations, have emerged as a pivotal technology for gene discovery, equipping clinicians with the ability to identify novel mutations [12,13]. Consequently, this has resulted in a notable rise in the detection of previously unknown mutations [14], which are frequently encountered in pediatric neurology clinics in the form of genetic and inherited diseases. Nonetheless, this study presented evidence of a growing number of pediatric neurologists in training in comparison to prior literature. A shift in demographic data was also remarkable, as previous studies highlighted a major predominance of men in this field, i.e., reaching a percentage as high as 73% [8], while our study estimated percentages of men and women of 46% and 54%, respectively. This remarkable change is consistent with the trend of medical specialties becoming more equitable, even in conservative communities [15,16,17]. Despite the fact that more than two-thirds of participants were married (68.29%), 63.41% of participants had no form of health insurance. This could be attributed to the current reality of a universal healthcare model adopted in Saudi Arabia which provides free access to health services to all citizens [18]. However, this finding will likely change in the future, as the country shifts toward privatizing the health sector, thereby increasing the number of health insurance holders [19]. This issue is of great importance, as studies have identified a correlation between insurance status and health [20]; this is often reflected in patients in terms of the quality of services provided by their healthcare provider.

The vast majority of participants were locally trained, despite the nationwide trend of expanding residency programs through international, government-funded scholarships. The majority, nonetheless, were either trained or were practicing in a main city, leaving peripheral cities and provinces in need of intensive planning and strategies to ensure fair distribution of programs and care providers. The engagement in clinical pediatric neurology research remains low. This issue was echoed among Saudi trainees in multiple other medical specialties [21], despite an increase in research resources and newly introduced tools including artificial intelligent software. Such tools, if adopted by trainees, can potentially advance their work in the field [22]. EEG and epilepsy were also the most prevalent types of fellowship training.

Moreover, a concerning point arose from the current data, i.e., that nearly half of the participants (48%) experienced some form of delay in acquiring their first job after graduating from their residency program, while others were either still in training or obtained an job immediately following their graduation. The issue is that nearly 26% of respondents did not obtain their first job until 4–6 months after graduating, with some experiencing multiple years of unemployment (5%). The struggle remains due to job listings for which pediatric neurologists continue to compete with general pediatricians in Saudi Arabia. This issue that could be addressed if independent job vacancies were offered specifically to pediatric neurologists. Furthermore, in our previous workforce analysis, it was found that nearly 18% of pediatric neurologists reported seeing >20 patients per half-day session [8]; in the present study, this figure has increased to 32% (Figure 3). As for supporting services, EEG remains the leading service in the current study, as in previous data (Figure 4) [8]; an increase in the availability of physical therapy is also reported [8].

Due to the study design (cross-sectional), the absence of some participants may have caused selection bias. The lack of sufficient updates and previous studies nationwide made it difficult to compare the findings. Unequal sampling may also be a potential limitation to the present study. The lack of census and public data on job types made it difficult to fully specify the participants’ job types. These limitations, however, were addressed by increasing the sample size and by detailed self-reporting.

## 5. Conclusions

In this study, our analysis led us to conclude that the demographic characteristics of the pediatric neurology workforce are undergoing significant transformations, particularly with an increasing number of female practitioners joining the field. A surge in women enrolling compared with previous studies was observed. However, it is crucial to note that the distribution of this workforce remains concentrated in limited areas across the country, primarily major cities, leaving peripheral cities in need of qualified professionals. The lack of well-trained and qualified pediatric neurologists in some cities constitutes a major challenge to the delivery of optimal services there. Real-life challenges, such as the delay in securing employment after graduation, have been comprehensively described and analyzed in this study. However, further evaluation by stakeholders is necessary to fully comprehend the multifaceted impact of such issues. Addressing these critical challenges and issues which persist for pediatric neurologists should be prioritized. Enhancing employment opportunities and reducing staffing turnover are essential aspects that must be considered when formulating strategies in the field of pediatric neurology, especially in Saudi Arabia. Given the small size of the sample in this study, it is recommended that similar research and studies be undertaken and expanded to yield more detailed and comprehensive findings.

## Figures and Tables

**Figure 1 healthcare-11-02288-f001:**
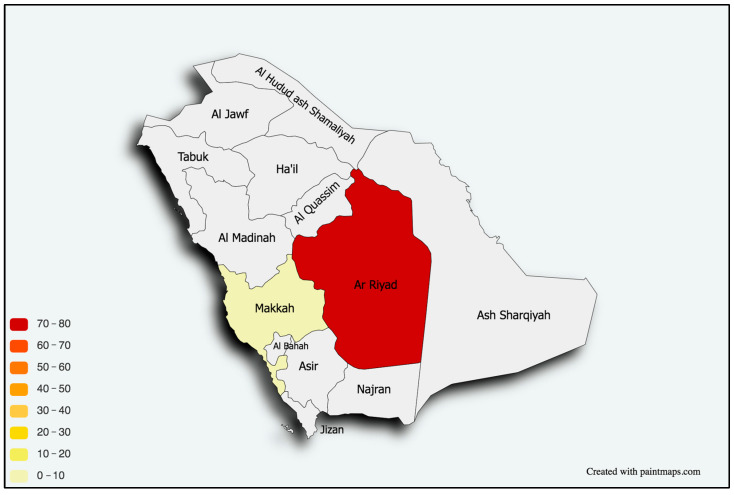
Distribution of pediatric neurologists in Saudi Arabia according to their current city of practice/training.

**Figure 2 healthcare-11-02288-f002:**
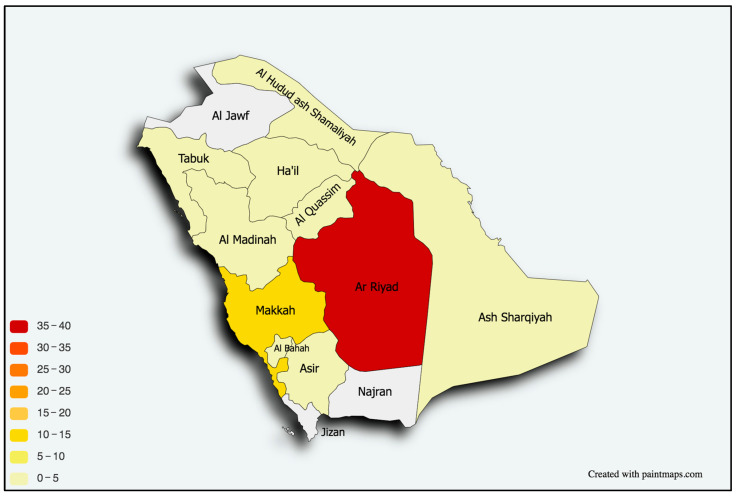
Distribution of pediatric neurologists in Saudi Arabia according to the city of their medical school.

**Figure 3 healthcare-11-02288-f003:**
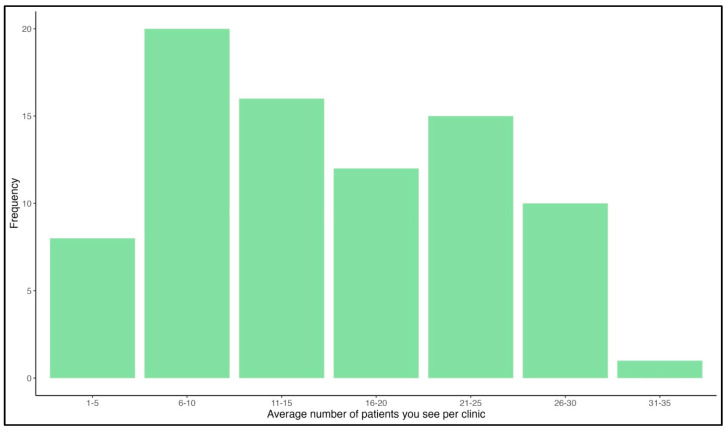
Average number of patients per half-day session.

**Figure 4 healthcare-11-02288-f004:**
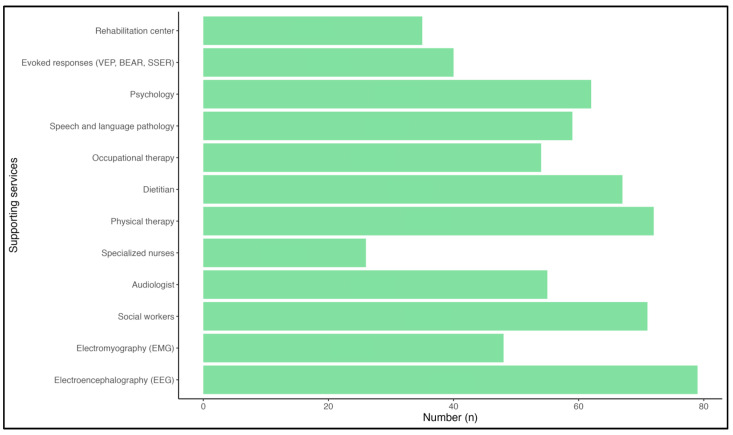
The available supporting services in respondents’ current center of practice. Abbreviations: VEP, visual evoked potential; BAER, brainstem auditory evoked response; SSER, somatosensory evoked response.

**Table 1 healthcare-11-02288-t001:** Descriptive demographic data.

Variable	Descriptive Statistic (n = 82)
Age mean (sd)	33.24 (5.66)
Gender n (%)	
Women	44 (53.66)
Men	38 (46.34)
Nationality n (%)	
Saudi	81 (98.78)
Relationship status n (%)	
Single	24 (29.27)
Engaged	2 (2.44)
Married	56 (68.29)
Current training levels n (%)	
Recent graduate	16 (19.51)
Postgraduate studies	39 (47.56)
R1	2 (2.44)
R2	1 (1.22)
R3	12 (14.63)
R4	6 (7.32)
R5	6 (7.32)
Monthly salary n (%) (1$ = 3.75 SAR)	
>15,000 SAR	49 (59.76)
12,000–15,000 SAR	2 (2.44)
<1000 SAR	2 (2.44)
NA	29 (35.37)
Health insurance n (%)	
No	52 (63.41)
Yes	30 (36.59)

Abbreviation: n: number of responders; R: Residency year; SAR: Saudi Arabian Riyal (1$ = 3.75 SAR); NA: Not answered.

**Table 2 healthcare-11-02288-t002:** Education, research, and training.

Variable	Descriptive Statistic (n = 82)
Country of residency n (%)	
Saudi Arabia	80 (97.56)
Residency center n (%)	
Alnoor Specialist Hospital	1 (1.22)
KAUH	3 (3.66)
KFMC	29 (35.37)
KFSHRC—Jeddah	1 (1.22)
KFSHRC—Riyadh	15 (18.29)
KAMC	3 (3.66)
KKUH	2 (2.44)
KSMC	4 (4.88)
NGHA	9 (10.98)
PSMMC	12 (14.63)
Residency start year n (%)	
1999–2003	2 (2.44)
2004–2008	4 (4.88)
2009–2013	10 (12.20)
2014–2018	44 (53.66)
2019–2023	22 (26.83)
Residency graduation year n (%)	
2004–2008	2 (2.44)
2009–2013	2 (2.44)
2014–2018	12 (14.63)
2019–2023	44 (53.66)
2024–2028	22 (26.83)
Number of research publications $\bar{x}$ (sd)	
2.41 (1.76)	82 (100)
Current fellowship enrollment n (%)	
No	58 (70.73)
Yes	24 (29.27)
Fellowship training n (%)	
Clinical neurophysiology	3 (3.66)
Epilepsy/EEG	13 (15.85)
Neurogenetics	1 (1.22)
Neuroimmunology	1 (1.22)
Neuromuscular/EMG	2 (2.44)
Other	4 (4.88)
N/A	58 (70.73)
Fellowship country n (%)	
Saudi Arabia	11 (13.41)
United States/Canada/Australia	13 (15.85)
N/A	58 (70.73)

Abbreviation: n: number of responders; KAUH: King Abdulaziz University Hospital; KFMC: King Fahad Medical City; KFSHRC: King Faisal Specialist Hospital; KAMC: King Abdulaziz Medical City; KKUH: King Khalid University Hospital; KSMC: King Saud Medical City; NGHA: National Guard Health Affairs; PSMMC: Prince Sultan Military Medical City; EEG: electroencephalogram; EMG: Electromyography; $\bar{x}$: mean; s: standard deviation. We removed two observations regarding country of residency to maintain the privacy of respondents.

**Table 3 healthcare-11-02288-t003:** Employment status.

Variable	Descriptive Statistic (n = 82)
City of birth n (%)	
Abha	3 (4)
Alhassa	4 (5)
Al Ula	0 (0)
Al Zulfi	0 (0)
Arkansas—USA	0 (0)
Athens—Greece	0 (0)
Bishah	0 (0)
Dammam	0 (0)
Jazan	0 (0)
Jeddah	4 (5)
Madinah	3 (4)
Makkah	3 (4)
Riyadh	15 (18)
Sharourah	0 (0)
Taif	0 (0)
Unaizah	0 (0)
Albaha	0 (0)
Months before first job n (%)	
1–3 months	18 (46.15)
4–6 months	8 (20.51)
7–9 months	3 (7.69)
9–12 months	6 (15.38)
1–3 years	3 (7.69)
6–10 years	1 (2.56)
City of first job n (%)	
Abha	3 (3.66)
Alhassa	4 (4.88)
Jeddah	8 (9.76)
Madinah	4 (4.88)
Makkah	5 (6.10)
Riyadh	40 (48.78)
Tabuk	2 (2.44)
N/A	10 (12.20)
Average number of patients per clinic n (%)	
1–5	8 (9.76)
6–10	20 (24.39)
11–15	16 (19.51)
16–20	12 (14.63)
21–25	15 (18.29)
26–30	10 (12.20)
31–35	1 (1.22)

**Table 4 healthcare-11-02288-t004:** Bivariate analysis to correlate between two groups and multiple variables.

Characteristic	Overall (n = 39)	Group 1 (n = 18)	Group 2 (n = 21)	*p*-Value
Age mean (SD)	33.77 (3.19)	33.94 (4.04)	33.62 (2.33)	0.97
Gender n (%)				1.00
Women	21 (53.85)	10 (55.56)	11 (52.38)	
Men	18 (46.15)	8 (44.44)	10 (47.62)	
City of birth n (%)				0.13
Abha	4 (10.26)	3 (16.67)	1 (4.76)	
Alhassa	2 (5.13)	2 (11.11)	-	
Arar	1 (2.56)	1 (5.56)	-	
Bishah	1 (2.56)	1 (5.56)	-	
Dammam	2 (5.13)	-	2 (9.52)	
Hail	1 (2.56)	1 (5.56)	-	
Jazan	3 (7.69)	-	3 (14.29)	
Jeddah	6 (15.38)	4 (22.22)	2 (9.52)	
Madinah	5 (12.82)	1 (5.56)	4 (19.05)	
Makkah	3 (7.69)	2 (11.11)	1 (4.76)	
Riyadh	8 (20.51)	2 (11.11)	6 (28.57)	
Tabuk	2 (5.13)	1 (5.56)	1 (4.76)	
Unaizah	1 (2.56)	-	1 (4.76)	
Residency center n (%)				S: 0.04
KFMC	16 (41.03)	5 (27.78)	11 (52.38)	
KFSHRC—Jeddah	1 (2.56)	1 (5.56)	-	
KFSHRC—Riyadh	7 (17.95)	3 (16.67)	4 (19.05)	
KKUH	1 (2.56)	1 (5.56)	-	
KSMC	1 (2.56)	1 (5.56)	-	
NGHA	6 (15.38)	1 (5.56)	5 (23.81)	
PSMMC	5 (12.82)	5 (27.78)	-	
University of Toronto	1 (2.56)	1 (5.56)	-	

Abbreviations: S: significant *p*-value; KFMC: King Fahad Medical City; KFSHRC: King Faisal Specialist Hospital; KKUH: King Khalid University Hospital; KSMC: King Saud Medical City; NGHA: National Guard Health Affairs; PSMMC: Prince Sultan Military Medical City. Group 1 represents responders who obtained a job in <3 months after graduation. Group 2 represents responders who obtained a job in 3 or more months after graduation.

## Data Availability

The datasets generated or analyzed during the current study are available from the corresponding author upon reasonable request.

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
