# Peer review of "Pediatric Neurology Workforce in Saudi Arabia: A 5-Year Update"

_healthcare, 2023, doi:10.3390/healthcare11162288_

Round 1

Reviewer 1 Report

Thank you very much for give me the opportunity to review this interesting paper titled Pediatric Neurology Workforce in Saudi Arabia: A 5-Year Up-2 date, that analyzed the current challenges faced by the pediatric neurology workforce through a community-based exploratory study among 82 practicing pediatric neurologists 28 as well as pediatric neurology trainees across the 13 regions of Saudi Arabia. As the authors note, the field of child neurology has undergone remarkable transformations in recent years, with significant challenges emerging for current and future pediatric neurologists that require careful consideration and proactive measures to address them.

Altogether it is an interesting study quite well written, and its objective is well defined.

Say this, here are a few comments:

1.     I am not very sure of the titles chosen for the headings of section 2 of materials and methods. I am more used to “participants and procedure” than to study design and setting”. In 2.1. it would be helpful if you at least indicated the number of participants in your study. As for 2.2, that I understand refers to your measurements, I'm not quite sure what exactly you mean by validation.

2.     In my humble opinion, perhaps it would be interesting to restructure the discussion, for example, figure 4 seems more typical of a results section, adding a greater volume of studies related to the objective of yours with which to discuss. Maybe the discussion section might be improved, in example, also adding some more arguments regarding the limitations, practical implications and suggestions for future research of your work.

3.     Please check the format (for example line 117 “All”, line 148 “n n: number of responders)

4.     Please check the references section, it does not seem to completely follow the format indicated by the journal. Likewise, if possible, it could be interesting to incorporate a greater number of references, especially more recent ones (only one data from 2022 and only three of 21 from 2023).

Author Response

Dear reviewer, 

Thank you for your insightful notes. I attached a file containing an address to your valuable comments. Comments number 1,2,3 were addressed as you specified. Comment number 4 was also addressed and all references were reviewed and modified however I wanted to note that we used mendeley software in accordance to the journal requirements. Regarding outdated references, the issue is the topic lack recent publications and articles published in 2022 and 2023 (mostly post-covid period) are extremely scarce and therefore comes our article to address the issue.

Kind regards, 

Reviewer 2 Report

This is a cross-sectional analysis using a survey to present the distribution of pediatric neurologists in Saudi Arabia (SA). The study is well written, but there are a few minor and one major problem with this study.

The major problem with the study is the lack of an "objective," as described in the last paragraph of the discussion and notified in the conclusion; the study aims to determine how pediatric neurologists are equally distributed in SA. The authors were not able to address this question; here are a few suggestions to address this aim:

-         The authors need to find the distribution of children based on the region (reported in Fig 1 and Fig 2) and compute the number of specialists per 10,000  population under coverage and then look at the data to find how specialists are distributed to address the study population.

- Some of the neurologist services are center-based services. This is very normal to have a higher specialist in the capital due to highly equipped hospitals. Still, the question that needs to be discussed in the discussion section is what type of service (e.g., transportation, etc.) provides by insurance coverage to address the population's need to receive assistance in the capital or nearby cities.

Here are a few minor issues:

Abstract;

"Aim of the study: This study aims to analyze the current challenges faced by the pediatric neurology workforce.", this is not a study aim; please revise based on my above comment.

"Methods: In May 2023, a community-based exploratory study was conducted involving 82 practicing pediatric neurologists and pediatric neurology trainees across the 13 regions of Saudi Arabia. A comprehensive 26- 29 item questionnaire, divided into four sections, was utilized to assess the participants.", please modify it to: This is a cross-sectional analysis, using a survey to present the distribution of the pediatrics neurologist in Saudi Arabia (SA). The final analytical sample includes xxx in xx regions in SA. Descriptive analysis has been used to address the study question.

"Results: The 30 survey received responses from a total of 82 pediatric neurologists in Saudi Arabia (response rate 31 55.03%), with 38 (46.34%) being male and 44 (53.66%) being female. The mean age was 33±1.225, 32 years—the majority of participants practiced in major cities such as Riyadh and Jeddah. Around 33 two-thirds had published three or fewer research articles. Approximately 30% were enrolled in a 34 fellowship program. Nearly 50% of pediatric neurologists experienced some form of delay in obtaining their first job, ranging from 1 to 36 months.", this needs to be revised to address the study questions; see above comments.

Introduction:

Please provide some information about the population and number of pediatric neurologists and compare them with the developed countries or countries with a similar GDP on health to better understand the issue in SA.

Materials and Methods:

Please add the study population section and describe how many surveys have been collected, how many have been removed because of missing observations, and the final analytical samples.

Results:

-         To keep the privacy of the respondents, remove the "and only one (1%) was an Egyptian national", and remove the nationality from the T1.

-         Table 1, define R1-R5.

-         Table1, add a note to define 1$=?SAR

-         Table 2:

-         To keep the privacy of participants. Remove Canada and Australia from the table and just report SA.

-         Remove the undergraduate medical school from the table if authors prefer to make it as a table note.

-         Report the number of publications by mean/SD

-         Avoid reporting observation with 1-2, instead combine Neurologists/neuroimmunology, EMG as one category

-         Combine Australia, Canada, and the US as one category.

Figures I would show Fig 1 and then Fig 2.

-         Table 3, I keep Months before the first job and the number of patients.

- In Figure 3, I expected to see 6-10 as the 2nd bar. Can you have a similar Fig to show the patients/10000 population to normalize the data?

-        What is the value of Table 4? What is the main point of this table? I don't see any significant differences.

Author Response

Dear reviewer, 

Thank you for your valuable comments. Kindly find our responses.

This is a cross-sectional analysis using a survey to present the distribution of pediatric neurologists in Saudi Arabia (SA). The study is well written, but there are a few minor and one major problem with this study.

The major problem with the study is the lack of an "objective," as described in the last paragraph of the discussion and notified in the conclusion; the study aims to determine how pediatric neurologists are equally distributed in SA. The authors were not able to address this question; here are a few suggestions to address this aim:

-         The authors need to find the distribution of children based on the region (reported in Fig 1 and Fig 2) and compute the number of specialists per 10,000  population under coverage and then look at the data to find how specialists are distributed to address the study population.

During our initial drafting of the manuscript, computing child neurologists according to the distribution of children based on the region was planned. We aimed to do so especially since the previous study by Al-Nahdi et al. discussed it and we wanted to establish a chronologic relationship. However, we couldn't obtain an accurate data of the children distribution based on the region and you can verify using the Saudi statistics data from this link https://database.stats.gov.sa/home/indicator/535. However, we utilized this source in reference [5] according to the best interest of the study.

- Some of the neurologist services are center-based services. This is very normal to have a higher specialist in the capital due to highly equipped hospitals. Still, the question that needs to be discussed in the discussion section is what type of service (e.g., transportation, etc.) provides by insurance coverage to address the population's need to receive assistance in the capital or nearby cities.

Thank you for your comment. If we intended to address every single issue and associated services including transportation this study would divert from its clinical perspective. However, we could do such study in the near future.

Here are a few minor issues:

Abstract;

"Aim of the study: This study aims to analyze the current challenges faced by the pediatric neurology workforce.", this is not a study aim; please revise based on my above comment.

Thank you. We modified it accordingly.

"Methods: In May 2023, a community-based exploratory study was conducted involving 82 practicing pediatric neurologists and pediatric neurology trainees across the 13 regions of Saudi Arabia. A comprehensive 26- 29 item questionnaire, divided into four sections, was utilized to assess the participants.", please modify it to: This is a cross-sectional analysis, using a survey to present the distribution of the pediatrics neurologist in Saudi Arabia (SA). The final analytical sample includes xxx in xx regions in SA. Descriptive analysis has been used to address the study question.

Thank you for your comment. We added your suggestions and updated the methods in the abstracts accordingly.

We also added your suggesting of "This is a cross-sectional analysis, using a survey to present the distribution of the pediatric neurologist in Saudi Arabia (SA)." To the main methodology. However, we couldn't entirely replace this sentence with the previous statement "In May 2023, a community-based exploratory study was conducted involving 82 practicing pediatric neurologists and pediatric neurology trainees across the 13 regions of Saudi Arabia. A comprehensive 26- 29 item questionnaire, divided into four sections, was utilized to assess the participants." As this would lead to significant ambiguity on the tools we used which will no longer be mentioned. Regarding the second half of your comment we indeed mentioned the number of participants and their respective region in the methods section 2.1.

"Results: The 30 survey received responses from a total of 82 pediatric neurologists in Saudi Arabia (response rate 31 55.03%), with 38 (46.34%) being male and 44 (53.66%) being female. The mean age was 33±1.225, 32 years—the majority of participants practiced in major cities such as Riyadh and Jeddah. Around 33 two-thirds had published three or fewer research articles. Approximately 30% were enrolled in a 34 fellowship program. Nearly 50% of pediatric neurologists experienced some form of delay in obtaining their first job, ranging from 1 to 36 months.", this needs to be revised to address the study questions; see above comments.

Thank you for your comment. We rephrased and updated the study aim according to your explanation where we indicated that we will emphasize on the participants' characteristics and data. This means the results will be better representing the aim of the study.

Introduction:

Please provide some information about the population and number of pediatric neurologists and compare them with the developed countries or countries with a similar GDP on health to better understand the issue in SA.

Thank you for your comment. According to the most recent World Bank data, Switzerland, Turkey, and the Netherlands are the immediate 3 countries with GDP lower than Saudi Arabia and Indonesia, Spain, and Mexico are the immediate 3 countries with higher GDP. Roughly the difference is 200,000,000$ +-. Among the six countries, all non-European countries have absolutely no similar studies. Regarding European states. There's indeed one recent study which can be accessed in this link https://www.sciencedirect.com/science/article/pii/S1090379820301574 that address a different perspective of the workforce. However, as for Spain, there were some studies for pediatric neurologists but addresses different aspects including this study which discusses genetic testing among Spanish pediatric neurologists https://pubmed.ncbi.nlm.nih.gov/27890788/ and this study which we believe might be similar to ours https://pubmed.ncbi.nlm.nih.gov/15122538/ but unfortunately it was available in Spanish language only. For that we intensively compared our study we 2 previous Saudi studies.

Materials and Methods:

Please add the study population section and describe how many surveys have been collected, how many have been removed because of missing observations, and the final analytical samples.

Thank you for your comment. We did not exclude any survey as we carefully reached all possible pediatric neurologists and they filled their information accurately. The total number of participants is a reflection to the number of collected surveys. And the total number distributed was mentioned in the results as a response rate. If this was not satisfying we will be pleased to add a population section in the methods.

Results:

-         To keep the privacy of the respondents, remove the "and only one (1%) was an Egyptian national", and remove the nationality from the T1.

Noted and removed.

-         Table 1, define R1-R5.

Noted and defined.

-         Table1, add a note to define 1$=?SAR

Noted and defined.

-         Table 2:

-         To keep the privacy of participants. Remove Canada and Australia from the table and just report SA.

Noted and removed.

-         Remove the undergraduate medical school from the table if authors prefer to make it as a table note.

Noted and removed despite we believe its importance to the readers.

-         Report the number of publications by mean/SD

Updated.

  •         Avoid reporting observation with 1-2, instead combine Neurologists/neuroimmunology, EMG as one category

Thank you for your comment. Given that these are considered different type of fellowship programs, we decided to report it separately. This would be beneficial for future study to compare trends of fellowship enrollment among pediatric neurologists in Saudi Arabia.

-         Combine Australia, Canada, and the US as one category.

Modified. However, for the same reasons we mentioned in the previous comments; we believe they should be kept separated to observe future trends.

Figures I would show Fig 1 and then Fig 2.

-         Table 3, I keep Months before the first job and the number of patients.

Modified.

- In Figure 3, I expected to see 6-10 as the 2nd bar. Can you have a similar Fig to show the patients/10000 population to normalize the data?

This figure was created as a comparison point for the previously published study. We utilized similar graph to obtain better observations. We updated the 2nd bar issue.

-        What is the value of Table 4? What is the main point of this table? I don't see any significant differences.

Despite there were no significant differences the table was extremely important as it compares two groups. Those who had a job early after their graduation and those who had a job in a later period. This's commonly discussed among pediatric neurologists that some fresh graduated might experience difficulties, including male vs females, and according to their residency center. However, in this table we demonstrated no significant differences among those different groups.

Reviewer 3 Report

Review:            

Healthcare – ID healthcare-2511081

Title:                  

Pediatric Neurology Workforce in Saudi Arabia: A 5-Year Update

General comments:

1.      Overall: Interesting and worthwhile topic updating workforce planning, demographic change and rate of publications of specialists. The maps in Figs 1 and 2 are very good. I wonder if some of the table that are long lists of 1 or 2 individuals from different location could be condensed into dots on the maps sized according to the numbers and dispensing with some of the Tables. I can understand for the individual players in the national sector these are of interest, but why not a hose journal than rather than expect a broader academic audience to be interested in this very specific detail? Also, the overly detailed paragraphs with intricate information prior to the tables is repetitive and not especially informative for a reader trying to see what is important and why should they read this. Finally, below here I have commented at several points for you to be consistent with reporting numbers/decimal places. Too much exactness does not necessarily add value to readers., particularly when they vary from one part of the paper to another. These things are pretty fixable.

2.      Opening paragraphs need idiomatic edit for clarity in what is being said and how it is said to reader. Detailed comments in the list below.

3.      Efficient use of Google Forms and What’s up. Well done. In my jurisdiction this would require institutional ethics permission.  Was this done here, or is the editor okay that it was not required?  I think a comment one way or another would guide readers, placed in the first para of 2.2.

4.      General statistics like response rates and mean years best to appear as one decimal point. NO value added with more decimal places. Line 125 suddenly pops in a two-decimal figure – be consistent in reporting.

5.      Gender – many people would accept your female and male categories in Table 1 (from line129). I recommend since this is not a biology assessment but a social/workforce study that you instead use “Women” and “Men” I am strong on this point in various reviews I do – Male and Female are inappropriate terminologies borrowed from western medicine and psychologies; outdate conceptualising when it is not needed. Same change could be made to Table 4.

6.      Table 1 centring of right-hand column not helpful to reader. Line the decimal points up above one another. You nearly do this in Table 2 – why not make that exactly correct?  Below 3.2 heading the reported figures go back to whole numbers – the question of consistency in your reporting again. But in Table 4 the percentages are largely meaningless as is the decimal point to 2 places, It becomes just clutter – what is a reader supposed to learn from the Table? Write that briefly below and simplify the table is the general rule.

7.      Discussion section. This very long section in only two paragraphs should have the first paragraph cut into multiple paragraphs. In the second short limitations comment, perhaps you could briefly comment why a full specification of job types could not be obtained from Census data (none?) or government statistics department?

Specific points:

Place (line)

Suggestion/Comment

Line 28 and line 31

I don’t quite understand the repetition of the count 82, yet at line 32 is about half, ie 55.03%

Lines 32-35

Second part of abstract - There are a lot of very specific stats here – not sure that this much detail is appropriate at this point in the ms.

Line 47-8

Suggest different word that “Subsequently” – this does not follow from the previous sentence.

In the next line the word “Moreover” is not needed, either.

Line 56

“aspects” is not clear – needs re-writing.

Line 64

“way less” is colloquial, what about “way below” or “well below”?

Line 65

Suggest adding “greater” before the word “shortage”

Line 67

“scarcity” is not idiomatic English – suggest editing.

Line 77

“region” or “regions”?

Line 83

Should the word “a” be inserted before “few items”?

Line 92

Looks like “include” should be “included” – a general need for editing across the script.

Line 122

Express the year to only one decimal point (Go back and do the same in the introduction) – a mathematical exactness does not help the flow of the ms.

Line 122

“single” not “singles” – the later has a social meaning not wanted here.

Line 148

Do you mean to have to single “n” after the word “Abbreviation” that starts this line?

Dear authors

The introduction is the section most in need of improving the English idiomatic style. I have commented on these to assist with this work. Reviewer

Author Response

Dear reviewer, 

Thank you for your valuable comments.

Kindly find attached our response to your queries.

General comments:

  1. Overall: Interesting and worthwhile topic updating workforce planning, demographic change and rate of publications of specialists. The maps in Figs 1 and 2 are very good. I wonder if some of the table that are long lists of 1 or 2 individuals from different location could be condensed into dots on the maps sized according to the numbers and dispensing with some of the Tables. I can understand for the individual players in the national sector these are of interest, but why not a hose journal than rather than expect a broader academic audience to be interested in this very specific detail? Also, the overly detailed paragraphs with intricate information prior to the tables is repetitive and not especially informative for a reader trying to see what is important and why should they read this. Finally, below here I have commented at several points for you to be consistent with reporting numbers/decimal places. Too much exactness does not necessarily add value to readers., particularly when they vary from one part of the paper to another. These things are pretty fixable.

Thank you so much for your valuable explanation. The lengthy table data was something that was mentioned by another reviewer, we summarized and removed some variables accordingly.

  1. Opening paragraphs need idiomatic edit for clarity in what is being said and how it is said to reader. Detailed comments in the list below.

Thank you for your comment. Will look into the comments below.

  1. Efficient use of Google Forms and What’s up. Well done. In my jurisdiction this would require institutional ethics permission.  Was this done here, or is the editor okay that it was not required?  I think a comment one way or another would guide readers, placed in the first para of 2.2.

Thank you for your comment. Institutional ethics and all associated permissions were obtained. They were addressed during our initial submission with the journal and it was deemed appropriate.

  1. General statistics like response rates and mean years best to appear as one decimal point. NO value added with more decimal places. Line 125 suddenly pops in a two-decimal figure – be consistent in reporting.

Thank you for your comment. We just updated all figures in the text into a one decimal point. Percentages in the tables were kept at two-decimal. We hope it's more consistent now.

  1. Gender – many people would accept your female and male categories in Table 1 (from line129). I recommend since this is not a biology assessment but a social/workforce study that you instead use “Women” and “Men” I am strong on this point in various reviews I do – Male and Female are inappropriate terminologies borrowed from western medicine and psychologies; outdate conceptualising when it is not needed. Same change could be made to Table 4.

Thank you for your comment. I see and respect your point of view and I just updated it into "Women" and "Men" throughout the manuscript.

  1. Table 1 centring of right-hand column not helpful to reader. Line the decimal points up above one another. You nearly do this in Table 2 – why not make that exactlycorrect?  Below 3.2 heading the reported figures go back to whole numbers – the question of consistency in your reporting again. But in Table 4 the percentages are largely meaningless as is the decimal point to 2 places, It becomes just clutter – what is a reader supposed to learn from the Table? Write that briefly below and simplify the table is the general rule.

Thank for your comment. The tables, centering and alignment of the decimal points were made according to the journal's template and instructions. Regarding consistency in 3.2 I updated it according to your previous comment. Simplifying tables were also conducted as another reviewer suggested the removal and modification to some tables and were made accordingly.

  1. Discussion section. This very long section in only two paragraphs should have the first paragraph cut into multiple paragraphs. In the second short limitations comment, perhaps you could briefly comment why a full specification of job types could not be obtained from Census data (none?) or government statistics department?

Regarding cutting off the discussion into multiple paragraphs, it is now divided. Regarding the limitation of the full specification of jobs; I added it to the limitation. It was indeed the lack of public census that hindered reaching such information.

Specific points:

Place (line)

Suggestion/Comment

Line 28 and line 31

I don’t quite understand the repetition of the count 82, yet at line 32 is about half, ie 55.03%

Modified.

Lines 32-35

Second part of abstract - There are a lot of very specific stats here – not sure that this much detail is appropriate at this point in the ms.

Modified by deleting two sentences to keep it concise.

Line 47-8

Suggest different word that “Subsequently” – this does not follow from the previous sentence.

In the next line the word “Moreover” is not needed, either.

Modified and "moreover" was removed.

Line 56

“aspects” is not clear – needs re-writing.

Noted.

Line 64

“way less” is colloquial, what about “way below” or “well below”?

Modified into well below.

Line 65

Suggest adding “greater” before the word “shortage”

"greater" added.

Line 67

“scarcity” is not idiomatic English – suggest editing.

Changed into "lack of"

Line 77

“region” or “regions”?

Noted and modified.

Line 83

Should the word “a” be inserted before “few items”?

Added.

Line 92

Looks like “include” should be “included” – a general need for editing across the script.

All uniformed into "included"

Line 122

Express the year to only one decimal point (Go back and do the same in the introduction) – a mathematical exactness does not help the flow of the ms.

In another comment by a reviewer we were asked to extensively summarize the figures and according to the journal's general instructions. For that we adopted such decimal style for years. However the decimal point for percentages were unified according to your comment.

Line 122

“single” not “singles” – the later has a social meaning not wanted here.

Changed to single.

Line 148

Do you mean to have to single “n” after the word “Abbreviation” that starts this line?

Yes that was what we intended as all abbreviation needs to be defined as per the submission requirements.

Round 2

Reviewer 2 Report

Thank you for addressing my comments, please remove n=1 from all tables e.g.:

T2, Canada and Australia and add a note that "we removed xx number of observations from some specific cities to keep the privacy of respondents."

T3, Arar, Hail, Tabuk, etc. and add a note that "we removed xx number of observations from some specific cities to keep the privacy of respondents."

T4, a few with one obs. and add a note that "we removed xx number of observations from some specific cities to keep the privacy of respondents."

Author Response

Dear reviewer, 

It was our pleasure to have you peer-reviewing our article. Your comments were valuable and improved our work. Regarding the comments you mentioned in this round, we addressed all of them, all n=1 were removed and a note was added accordingly. However, in Table 4, we weren't able to remove n=1 as it may impact the calculated p=values in which the multi-linear regression was based collectively on all the participants within a single group. However in Table 4 the n=1 isn't revealing to participants data. If you still consider it a preech to privacy we would be happy to remove it and re-do the analysis to detect any changes in p=value.

Thank you once again, and please find attached the updated version. 

Kind regards,

Anas

Reviewer 3 Report

Thank you for taking the trouble to make considerable changes to your manuscript. This has considerably improved the flow and the expression. In the attached for my suggested I've notes dozen or so additional minor changes that you might make; otherwise I commend you for documenting their health needs in this aspect of Saudi Arabia.

Author Response

Dear reviewer, 

It was our pleasure to have you peer-reviewing our article. Your comments were valuable and improved our work.

Regarding the comments you mentioned in this round, we addressed all of them as follows: 

  1. We unified "health care" to "healthcare". I apologize as this error was made after adding those statements at the request of another reviewer. I double-checked it and it is now spelled "healthcare" throughout the manuscript.
  2. We replaced the "grow" with "increase" which was linguistically more suitable.
  3. Word modified into the past tense "continued". Again this was inconsistent due to the late addition of that paragraph. 
  4. It was deleted. Again this error occurred due to the late addition of that paragraph.
  5. A plural "S" was added to "shortage" as we were referring to multiple shortages. 
  6. Changed into "pediatric neurologists"
  7. Changed into "a"
  8. Specialist changed into "specialists"
  9. Changed to supporting
  10. The sentence changed to "a major predominance of men in the speciality"
  11. We changed to "women enrolling"

Thank you once again, and please find attached the updated version. We also went through the other part of the manuscript and revised it again by our co-authors so we hope no further linguistic issues are present and thus the work qualifies for "English language fine. No issues detected" in the Quality of English Language assessment by you. 

We sincerely appreciate your efforts.

Kind regards,

Anas
